# The Novel Cerato-Platanin-Like Protein FocCP1 from *Fusarium oxysporum* Triggers an Immune Response in Plants

**DOI:** 10.3390/ijms20112849

**Published:** 2019-06-11

**Authors:** Songwei Li, Yijie Dong, Lin Li, Yi Zhang, Xiufen Yang, Hongmei Zeng, Mingwang Shi, Xinwu Pei, Dewen Qiu, Qianhua Yuan

**Affiliations:** 1Hainan Key Laboratory for Sustainable Utilization of Tropical Bio-resources, Institute of Tropical Agriculture and Forestry, Hainan University, Haikou 570228, China; lear9999@163.com; 2The State Key Laboratory for Biology of Plant Diseases and Insect Pests, Institute of Plant Protection, Chinese Academy of Agricultural Sciences, Beijing 100081, China; dongyijie8837@163.com (Y.D.); cherryli@live.cn (L.L.); zhangyi825@outlook.com (Y.Z.); yangxiufen@caas.cn (X.Y.); zenghongmei@caas.cn (H.Z.); 3The School of Resources and Environment Science, Henan Institute of Science and Technology, Xinxiang 453003, China; Shimw888@126.com; 4Ministry of Agriculture Key Laboratory on Safety Assessment (Molecular) of Agriculture Genetically Modified Organisms, Biotechnology Research Institute, Chinese Academy of Agriculture Sciences, Beijing 100081, China; peixinwu@caas.cn

**Keywords:** *Fusarium oxysporum*, cerato-platanin, elicitor, FocCP1, immune response, systemic acquired resistance

## Abstract

Panama disease, or Fusarium wilt, the most serious disease in banana cultivation, is caused by *Fusarium oxysporum f.* sp. *cubense* (FOC) and has led to great economic losses worldwide. One effective way to combat this disease is by enhancing host plant resistance. The cerato-platanin protein (CPP) family is a group of small secreted cysteine-rich proteins in filamentous fungi. CPPs as elicitors can trigger the immune system resulting in defense responses in plants. In this study, we characterized a novel cerato-platanin-like protein in the secretome of *Fusarium oxysporum f.* sp. *cubense* race 4 (FOC4), named FocCP1. In tobacco, the purified recombinant FocCP1 protein caused accumulation of reactive oxygen species (ROS), formation of necrotic reaction, deposition of callose, expression of defense-related genes, and accumulation of salicylic acid (SA) and jasmonic acid (JA) in tobacco. These results indicated that FocCP1 triggered a hypersensitive response (HR) and systemic acquired resistance (SAR) in tobacco. Furthermore, FocCP1 enhanced resistance tobacco mosaic virus (TMV) disease and *Pseudomonas syringae* pv. *tabaci* 6605 (*Pst.* 6605) infection in tobacco and improved banana seedling resistance to FOC4. All results provide the possibility of further research on immune mechanisms of plant and pathogen interactions, and lay a foundation for a new biological strategy of banana wilt control in the future.

## 1. Introduction

The filamentous fungus *Fusarium oxysporum f.* sp. *cubense* (FOC) is the causal agent of banana fusarium wilt or Panama disease. Four races (1, 2, 3, and 4) of FOC have been identified based on their pathogenicity to different banana varieties. Race 1 and race 2 infect *GrosMichel* and *Bluggoe* banana cultivars, respectively. Race 3 attacks *Heliconia* sp. and has a slight effect on banana. Race 4, the most serious pathogenic race, attacks *Mimosa nana Lour* and *Musa basjoo Siebold*, and causes serious banana wilt [1,2]. *Fusarium oxysporum* is a worldwide soil-borne pathogen and its chlamydospore survives in host plants or soil for decades. When the environment is suitable, mycelia penetrate the banana root and secrete a gel causing a blockage in vascular vessels, blocking water transportation in the plant, then resulting in a reddish-brown discoloration and necrosis of the rhizome and pseudostem, eventually leading to leaf collapse and plant death [1,3].

Up until now, chemical control is still the main way to prevent and control banana blight. One of the most effective ways to combat disease is by enhancing host plant resistance [1]. Previous studies have reported that several toxin proteins and cell wall-degrading enzymes secreted by *Fusarium oxysporum* played a key role in the relationship between pathogen and plant. In addition, these proteins acted as effectors and stimulated plant defense responses [4,5]. Currently, there is still limited knowledge about the function of effectors or necrosis-inducing molecules in these pathogens.

The cerato-platanin protein (CPP) family was first described in culture filtrate of *Ceratocystis platani* [6], which is a necrotrophic fungus that infects plane trees [7]. Subsequently, more CPPs have been characterized in filamentous fungi, such as BcSpl1 from *Botrytis cinerea* [8], MpCP1 from *Moniliophthora perniciosa* [9], Snodprot1 from *Phaeosphaeria nodorum* [10], FgCPP1 and FgCPP2 from *Fusarium graminearum* [11], Sp1 from *Leptosphaeria maculans* [12], Sm1 from *Trichoderma virens*, Epl1 from *Trichoderma*, and VdCP1 from *Verticillium dahliae* [13,14,15]. Many of these CPPs were only identified from fungal fermentation medium, indicating that CPP is a secreted protein [6,16,17].

CPPs are highly conserved proteins, which consist of a 16–19 amino acid signal peptide and 120–134 amino acids with four cysteines forming two disulfide bridges [14,18,19]. Previous studies have demonstrated that a few CPPs exist in the fungal cell wall with a chitin and N-acetylglucosamine (GlcNac) oligomer binding site [20]. These CPPs play important roles during plant infection, and CPPs are also regarded as microbe-associated molecular patterns or virulence factors, which induced necrosis and promoted expression of defense genes in plants [21,22]. CPPs act as elicitors, inducing synthesis of reactive oxygen species (ROS) and a local hypersensitive response (HR) in plant leaves [4,6]. BcSpl1 from *Botrytis cinerea* induced systemic acquired resistance (SAR) in tobacco with regulation of salicylic acid (SA), indicating that the CP-mediated plant defense involved the SA signaling pathway [4]. However, no CPP has been characterized in *Fusarium oxysporum f.* sp. *cubense* race 4 (FOC4).

In order to respond to pathogen infection, plants have evolved two layers of sophisticated immune defense: PTI, pathogen-associated molecular pattern (PAMP)-triggered immunity; and ETI, effector-triggered immunity [17,23]. One of the main characteristics of PTI is that PAMPs possess some small and highly conserved amino acid sequences. Identified PAMPs mainly include chitin, bacterial flagellin, EF-Tu, LCOs, lipopolysaccharides, peptidoglycans, and oligogalacturonide. These PAMPs are monitored and identified by membrane-anchored pattern recognition receptors (PRRs) in plant cell membranes [24,25]. Pathogens have evolved effector molecules to cope with the PTI in host plants to colonize the plant. Pathogen effectors are recognized by one type of NB-LRR (binding and leucine-rich repeat nucleotide domains) proteins in plant cells; these proteins initiate the plant ETI immune response [25,26]. ETI is a rapid and strong immune response associated with production of ROS and subsequent HR [25,27]. However, PTI and ETI cannot be clearly distinguished due to similar PRRs [28]. Plants activate their own systemic immune responses depending on mutual recognition of pathogenic microorganisms. In this interaction process, plants regulate defense-related gene expression, trigger defense signal pathways, produce a variety of defensive substances, and cause HR and SAR to resist pathogen infection [26].

Here, we report a novel cerato-platanin-like protein in culture filtrate of FOC4, named FocCP1. The purified recombinant FocCP1 protein led to accumulation of H_2_O_2_ and HR in tobacco, and activated production of ROS in tobacco suspension cells. FocCP1 induced expression of defense-related genes and enhanced accumulation of hormones in tobacco. Moreover, we found that FocCP1 induced tobacco systemic defense responses and elicited banana plant responses against FOC4. All results provide new insights into studying the relationship between this pathogen and plant, and lay a foundation for a new biological strategy of banana wilt control in the future.

## 2. Results

### 2.1. Identification of Secreted Proteins

The extracted proteins were subjected to mass spectrographic analysis and we obtained 10,221 peptides and 2235 proteins in secretory proteomics of FOC4. All proteins were subjected to Gene Ontology (GO) functional annotation analysis; these proteins covered a wide range of molecular functions (MF), cellular components (CC), and biological processes (BP). MF included oxido-reductase activity, nucleotide binding, hydrolase activity, kinase activity; CC included membrane, integral component of membrane, and cytoplasm; and BP included metabolic process, oxidation-reduction process, translation, and proteolysis. The top 30 identified proteins of GO functional annotation analysis are listed in Appendix A. Some proteins potentially secreted as a result of the pathogen–host interaction were identified (Table 1), including cellulase, chitinase, carboxypeptidase, lipase, aminopeptidase, glucanase, and one cerato-platanin-like protein (UniProtKB: X0JM38).

### 2.2. FocCP1 is A Member of the Cerato-Platanin Family

We obtained homologous proteins of FocCP1 in the UniProt protein database (https://www.uniprot.org/uniprot, accessed on 25 March, 2019), and searched nucleic acid sequences in the NCBI GenBank database (https://www.ncbi.nlm.nih.gov/, accessed on 25 March, 2019). The open reading frame of FocCP1 contained a 420 bp nucleotide sequence of 139 amino acids with a predicted N-terminal SP (1–18 amino acids), suggesting that FocCP1 might be a secreted protein (Appendix A). The phylogenetic analysis indicated that CPPs were widely present in fungi, including several important plant pathogens. FocCP1 had high homology with other CPPs (Figure 1a). The predicted 3D structure demonstrated that FocCP1 had three α-helices and eight β-strands, which was highly similar to Foc1CP (protein SnodProt1 from *F. oxysporum f.* sp. *cubense* race 1: N4TJM2) and FolCP (uncharacterized protein from *F. oxysporum f.* sp. *lycopersici*: A0A0D2YA76) (Figure 1b), suggesting that FocCP1 might have an analogous function to other CPPs. Conserved motif searched by Multiple Em for Motif Elicitation (MEME) software (Motif-based sequence analysis tools) indicated that CPP proteins possessed three main conserved motifs (Figure 1c). These results show that FocCP1 is a member of the CPP family.

### 2.3. Cloning, Expression, and Purification of the Recombinant FocCP1 Protein

The full-length cDNA encoding FocCP1 was amplified from *F. oxysporum* by qRT-PCR. The open reading frame of FocCP1 without signal peptide or a stop codon was amplified by *pPICZαA-FocCP1*-F/*pPICZαA-FocCP1*-R primers. The recombinant *pPICZαA*-*FocCP1* plasmid was transformed into *Pichia pastoris* cells for protein expression (Appendix A). The recombinant FocCP1 protein was purified using an ÄKTA Protein Purifier System. Only one protein peak was present in the map under 280 nm ultraviolet (UV) light, indicating that the target FocCP1 protein was separated by affinity chromatography with a His-Trap HP mini-column and His-Trap desalting column (Figure 2a). A single protein band was verified by Coomassie blue R-250 staining in sodium dodecyl sulfate polyacrylamide gel electrophoresis (SDS-PAGE). Recombinant FocCP1 protein with an apparent molecular weight of approximately 14.58 kDa was successfully expressed and purified (Figure 2b).

### 2.4. FocCP1 Induced Hypersensitive Response (HR) and H_2_O_2_ Accumulation in Tobacco

To determine whether FocCP1 protein induced HR in tobacco, we observed that recombinant FocCP1 protein induced a systemic necrotic reaction in tobacco leaves; bovine serum albumin (BSA) did not cause a systemic necrotic reaction (Figure 3a). The DAB staining results showed that FocCP1 induced ROS at the infiltrated site in tobacco leaves, compared to BSA as a negative control and Flg22 as a positive control (Figure 3b). FocCP1 caused a rapid active oxygen burst in tobacco suspension cells (BY-2) after 5 min and reached a maximum at approximately 12–14 min. Subsequently, ROS gradually decreased to a similar level as that in the negative control (Figure 3c). These results showed that FocCP1 induced HR and ROS in tobacco.

We determined the minimum concentration of FocCP1 for HR activity in tobacco plants: 50 μM FocCP1 produced a weak necrotic spot, 75 μM produced necrotic lesions, and 100 μM resulted in significant necrotic lesions (Appendix A). We measured the stability of FocCP1 protein, and showed that FocCP1 had stable elicitor function below 70 °C and acid-base stability within pH 2–10 (Appendix A).

### 2.5. Defense Responses in Tobacco Caused by FocCP1

To test whether FocCP1 protein has a similar elicitor function in tobacco, we analyzed the expression of two pathogenesis-related (*PR*) genes, *PR-1* and *PR-5*, which are frequently used as marker genes for defense responses in plants. Expression of the *PR* genes was rapidly induced in elicitor-treated plants after 16 h and clearly induced after 24 h treatment, but there was no clear expression after 8 h (Figure 4a).

We also determined expression levels of genes involved in the salicylic acid (SA) and jasmonic acid (JA)/ethylene (Et) dependent defense pathway in tobacco plants. The SA signal-related genes (*PAL* and *EDS1*) were over-expressed after 16 and 24 h induction, but not obviously expressed after 8 h induction; the *PAL* gene was especially strongly induced, but strong induction was not observed for *EDS1.* The HR marker-related gene (*HSR203J*) and JA/Et signal-induced gene (*LOX*) were elevated in tobacco leaves treated with FocCP1 protein after 16 and 24 h induction (Figure 4a). The SA and JA/Et signaling pathways are considered the core pathway for plant defensive responses [29]. We also analyzed the expression levels of SA and JA in tobacco leaves induced by FocCP1. FocCP1 caused significant accumulation of SA and JA in tobacco leaves after 24 h treatment, compared to BSA treatment (Figure 4b).

### 2.6. Deposition of Callose and Phenolic Compounds in Tobacco Induced by FocCP1

In tobacco leaves stained with aniline blue, we observed a dot pattern of callose deposition at the mesophyll cell walls; Flg22 also induced callose deposition in the stained area, while BSA did not (Figure 5a,b). We determined whether FocCP1 protein induced phenolic compounds in tobacco suspended cells (BY-2). The results showed that FocCP1 and Flg22 treatment resulted in phenolic compounds in suspension cells. In contrast, BSA had no effect (Figure 5c,d). Secondary metabolic phenolic components are considered to enhance cell wall structure and act against pathogen infection; these phenolic compounds are often considered to be part of an active defense response associated with non-host resistance [30].

### 2.7. FocCP1 Triggers Plant Disease Resistance

The tobacco mosaic virus (TMV)-green fluorescent protein (GFP) lesions were observed and counted in tobacco leaves under ultraviolet light. The numbers and diameters of TMV lesions in FocCP1-treated plants were fewer and smaller than in the control plants (Figure 6a,b). Percent inhibition of TMV in tobacco leaves treated with FocCP1 reached 48.19%, while those treated with Flg22 reached 39.38%; these data indicated that the tobacco plants had enhanced resistance to TMV-GFP. The concentration of *P. syringae* pv. *tabaci* 6605 (*Pst.* 6605) in tobacco leaves treated with FocCP1 and Flg22 was significantly fewer than in BSA treated plants and the bacterial population had decreased by 30% at the fourth day. The results demonstrated that FocCP1 could enhance systemic resistance against *Pst.* 6605 (Figure 6c,d). In addition, we observed and counted brown bulbs and vascular bundles of banana seedlings. Compared to water and BSA, the incidence of banana wilt was significantly reduced: the banana plants showed lower degrees of leaf yellowing, vascular bundles and bulb browning (Figure 6e). According to the disease index analysis, FocCP1 triggered resistance of banana plants to FOC4. The inhibition rate of banana blight reached 40.63% (Figure 6f). In conclusion, all results suggested that FocCP1 induced resistance in tobacco against TMV and *P. s. tabaci* and enhanced banana seedling resistance against FOC4 infection.

## 3. Discussion

Secretory proteomics is an effective tool to explore fungus–plant interaction proteins [31,32]. We identified a few potentially secreted proteins through the proteome analysis of FOC4, including cellulase, chitinase, and glucanase. Secreted proteins from FOC4 may play an important role in infecting banana plants and provide entry points for studying plant and pathogen interactions. Among these secreted proteins, one secreted CPP, FocCP1, was identified to be widely present in filamentous fungi (Table 1). FocCP1 had high similarity and homology with other CP family proteins, a group of small secreted cysteine-rich proteins that are associated with the virulence of pathogenic fungi [18,33]. FocCP1 has three a-helices, eight β-strands, and three main conserved motifs similar to other CPPs (Figure 1). The proteomics and bioinformatics analysis indicated that FocCP1 was a novel, typical CPP from FOC4.

To verify the elicitor function of the FocCP1 protein, we constructed a eukaryotic expression vector and transformed it into *Pichia pastoris* cells for expression. The recombinant FocCP1 weighed approximately 14.58 kDa according to SDS-PAGE electrophoresis analysis (Figure 2). The main way to verify protein elicitor function is whether or not the protein induces necrosis and a significant hypersensitive reaction (HR) in tobacco [34]. Most CPPs, such as BcSpl from *Botrytis cinerea*, MgSM1 from *Magnaporthe grisea*, and Pop1 from *Ceratocystis populicola*, were able to cause necrosis in plants [6,16,35], while several CPPs, such as SP1 from *Leptosphaeria maculans* and Sm1 from *Trichoderma virens* did not induce cell death [12,13,36]. In this study, FocCP1 induced necrosis and HR in tobacco leaves.

ROS are involved in the early immune response in plants, and the primary ROS, including superoxide (O_2_^−^), hydrogen peroxide (H_2_O_2_), and nitric oxide (NO), lead to HR in plants [37,38], These signaling molecules are involved in defense response, and regulated signaling pathways through gene expression and metabolite changes [34]. ROS accumulation was observed in tobacco leaves and suspension cells treated by FocCP1 (Figure 3), showing that FocCP1 might be an effector secreted by *F. oxysporum*. Through temperature and acid-base stability detection, the FocCP1 protein had stable protein-stimulating activity. According to the minimum concentration detection, more than 50 μM FocCP1 caused necrotic lesions in tobacco leaves (Figure 4). We speculated that a low concentration of FocCP1 was not sufficient to trigger necrosis, possibly because tobacco plants were not sensitive to low concentrations of FocCP1, or FocCP1 only caused a weak HR in tobacco compared to Flg22.

Many elicitors have been isolated and identified from plant pathogens, and these elicitors have the ability to enhance plant resistance to pathogens, such as PeaT1 from *Alternaria tenuissima* and MoHrip1 from *Magnaporthe oryzae* [39,40,41]. In this study, FocCP1 induced up-regulation of defense-related genes, such as pathogenesis-related (*PR*) genes *PR1* and *PR5* (Figure 4). The up-regulation of *PR* genes involved the defense network in plants and enhanced plant ability to antagonize pathogens; all results showed that FocCP1 triggered an immune response in tobacco.

Research has confirmed that downstream responses were regulated by the balanced action of SA, JA, and Et [42,43]. *PAL* and *EDS1* genes play an important role in SA-signaling and basal defense, and *PAL* genes play an important role in biosynthesis of lignin, which creates a physical barrier for the infection of pathogens [44,45]. The *LOX* gene was considered part of the JA/Et pathway and it has been confirmed that JA-dependent responses are involved in plant defenses against necrotrophic pathogens [46,47]. In this study, *PAL*, *EDS1*, and *LOX* genes were up-regulated in tobacco after 16 h treatment, and accumulation of SA and JA was detected in tobacco leaves induced by FocCP1 after 24 h treatment (Figure 4). We speculated that the immune response in tobacco induced by FocCP1 protein involves the SA and JA pathways. In addition, the *HSR203J* gene is involved in chitinase synthesis and HR symptoms in plant [48]. In this study, *HSR203J* was up-regulated in tobacco leaves treated with FocCP1 after 16 h treatment, and HR symptoms were present in tobacco leaves after 24 h treatment (Figure 4). All results demonstrated that the elicitor FocCP1 induced a plant immune response mediating SA and JA signaling pathways, and caused hormone accumulation of SA and JA in tobacco. ROS is a key signal of the immune response in plant tissues: It triggers expression of resistance genes and callose deposition in plant cells [49,50]. ROS promote cell wall thickening, cause programmed cell death, induce defense gene expression, and enhance defense compound synthesis and secondary metabolite accumulation [51,52]. In this study, we observed callose deposition in tobacco leaves and accumulation of phenolic compounds in tobacco suspension cells induced by FocCP1 under UV fluorescence microscopy (Figure 5). Callose is a multifaceted defense response controlled by different signaling pathways, depending on environmental conditions and pathogen-associated molecular patterns [53]. Moreover, FocCP1 triggered tobacco resistance to TMV and *Pst.* 6605 and reduced the probability of *Fusarium oxysporum* infesting banana seedling (Figure 6). These results showed that FocCP1 triggered the plant immune response, caused secondary metabolite accumulation, and enhanced defense compound synthesis. More experiments should be proved that the FocCP1 induced SAR in banana.

Based on the above experimental results, we speculate that FocCP1 inhibits antagonistic activity through interaction with PRRs in plant cells; activates the plant immune response to improve disease resistance; regulates expression of *PRs*, *PAL*, *EDS1*, *LOX*, and *HSR203J*; adjusts the balance of SA and JA/Et; triggers chitinase synthesis and HR; induces callose and phenolic substances deposition; and lead to SAR in both host and non-host plants [54,55]. At present, the interaction mechanism between elicitor and plant is not very clearly, more experiments are required for verification.

## 4. Materials and Methods

### 4.1. Culture Conditions of Microorganisms and Plants

The high virulence strain FOC4 was conserved at the Environment and Plant Protection Institute, Chinese Academy of Tropical Agricultural Sciences. The fungus was cultured on potato dextrose agar medium (PDA) or in potato dextrose broth (PDB) at 25 °C in the dark. *Escherichia coli* (*E. coli*) DH5α and *E. coli* BL21 (DE3) cells were cultivated in Luria-Bertani (LB) agar medium or broth on a shaker at 37 °C. *Pst.* 6605 were stored in 20% glycerol at −80 °C and grown in a King’s B (KB) (K50 and Rif50) medium plate at 28 °C. Suspension cells of tobacco were cultured in MS medium in a shaker rotating at 130 rpm at 26 °C in the dark [41]. Tobacco plants (*Nicotiana benthamiana*) were grown in a greenhouse at 25 °C under a 16/8 h day/night cycle. Banana seedlings (*Musa paradisiaca* L. var. *sapientum* O. Ktze.) were planted in a growth chamber at 30 °C with day/night period of 16/8 h.

### 4.2. Label-Free Analysis of FOC4 Strain

FOC4 was cultured on a PDA plate for 5 days. The edge hyphae were picked into 100 mL of PDB liquid medium and cultured until the concentration of spores reached 10^6^/mL. Then, the FOC4 spore suspension was inoculated into PDB liquid medium (1 L) and cultured for 2 days. Subsequently, FOC4 fermentation liquid continued to be co-cultivated under inducing by hung sterilized banana seedlings for 2 days. The culture supernatant was centrifuged at 3500 g for 30 min at 4 °C and filtered with a 0.45 μm filter (Millipore, Suzhou, China). Total proteins in the supernatant were extracted with a trichloroacetic acid (TCA)-acetone precipitation method and dissolved in protein lysate buffer (8 M urea, 100 mM Tris-HCl, pH 8.0, 1 mM PMSF protease inhibitor) [56]. All proteins were analyzed with the high-performance liquid chromatography (HPLC) liquid phase system Ultimate 3000 (Thermo Scientific, Waltham, MA, USA); mass spectrometry was performed using a Q-Exactive HF mass spectrometer (Thermo Scientific, Waltham, MA, USA). The RAW files were identified and quantified using Mascot and Proteome Discoverer, and the results of filtration parameters were set as PSM FDR ≤0.01, Protein FDR ≤0.01 [57]. Label-free analysis of *F. oxysporum* was completed by Beijing Bio-Fly Bioscience Co., Ltd.

### 4.3. Bioinformatics Analysis

All reference sequences of CPPs were obtained using Blastp, nucleic acid sequences were searched in the NCBI (National Center for Biotechnology Information) GenBank database (https://www.ncbi.nlm.nih.gov/, accessed on 25 March, 2019), and amino acid sequences were from the UniProt protein database (https://www.uniprot.org/, accessed on 25 March, 2019). SignalP-4.1 (http://www.cbs.dtu.dk/services/SignalP/) was used to predict the signal peptide of FocCP1 [58]. The PHYRE 2 server (http://www.sbg.bio.ic.ac.uk/phyre2/, accessed on 2 April, 2019) with normal modeling mode was used to perform protein homology modeling. PYMOL 2.3 (https://pymol.org/2/) was used for editing the PDB file produced by Swiss-Pdb Viewer [59]. Muscle was used for generating amino acid sequence alignments (https://www.ebi.ac.uk/Tools/msa/muscle/) [60] and a phylogenetic tree was constructed in MEGA 6.0 (Molecular Evolutionary Genetics Analysis 6.0) using the maximum-likelihood method [61]. Sequence motifs of FocCP1 were identified and analyzed with the MEME 5.0.5 (http://meme-suite.org/) suite [62,63].

### 4.4. Cloning, Expression, and Purification of FocCP1

Total RNA was extracted with the E.Z.N.A^®^ Fungal RNA Kit (Omega, Norcross, GA, USA) according to the manufacturer’s protocols. First-strand cDNA was synthesized by TransScript^®^ II Reverse Transcriptase (TransGen Biotech, Beijing, China). The entire sequence of the *FocCP1* gene was amplified with PCR Super Mix II Kit (TransGen Biotech, Beijing, China), using the *FocCP1*-F/*FocCP1*-R primers listed in Table 2. The purified fragments were cloned into the pMD18-T Simple vector (TaKaRa, Dalian, China), transformed into *E. coli* DH5α cells (TransGen Biotech, Beijing, China), and then sequenced. FocCP1-encoding fragment lacking signal peptide and stop codon was amplified by *pPICZαA-FocCP1*-F/*pPICZαA-FocCP1*-R primers with *EcoR*I and *Xba*I sites (Table 2). The purified fragment was cloned into a pPICZαA (TransGen Biotech, Beijing, China) plasmid.

Subsequently, recombinant plasmid *pPICZαA*-*FocCP1* was linearized at PmeI sites and transformed into *Pichia pastoris* KM71H cells for expression [15]. Recombinant FocCP1 protein was separated in supernatant with an ÄKTA Explorer system (Amersham Biosciences, Boston, MC, USA) through affinity chromatography with a His-Trap HP mini-column (GE Healthcare, Waukesha, WI, USA) and a His-Trap desalting column (GE Healthcare, Waukesha, WI, USA). Purity and molecular masses of FocCP1 were analyzed by SDS-PAGE (12.5%) and the concentration was measured by BCA™ Protein Assay Kit (TransGen Biotech, Beijing, China). Purified FocCP1 protein was conserved in protein buffer (20 mM Tris-HCl, pH 8.0) at −80 °C.

### 4.5. Characteristics of FocCP1 Protein

To verify HR-inducing activity of FocCP1, 4-week-old tobacco leaves were infiltrated with purified 50 µL FocCP1 (100 μM) with a sterile 1 mL syringe for about 1 cm^2^. HR symptom necrosis was checked in the infiltrated areas after 24 h [41]; BSA (100 μM) and Flg22 (100 μM) acted as the negative and positive controls, respectively. To determine the suitable concentration of FocCP1 for HR, different concentrations (5, 10, 25, 50, 75, 100, and 200 μM) of FocCP1 solutions were infiltrated into tobacco leaves, with the same concentrations of BSA and Flg22 as the control. To measure protein heat stability, seven aliquots of the same concentration of FocCP1 (100 μM) were incubated at 25, 37, 50, 60, 70, 80, and 100 °C for 15 min, and HR activities of these proteins were tested as above. To detect acid-base equilibrium stability of the protein, FocCP1 (100 μM) was incubated in pH 1, 2, 4, 6, 7, 8, 10, and 12 for 20 min at 37 °C, then neutralized to pH 7; HR in tobacco leaves were determined as above [64]. In all experiments, 50 μL FocCP1 was infiltrated into tobacco leaves with approximately 2 cm^2^ area.

### 4.6. Detection of Hydrogen Peroxide in Tobacco

Four-week-old tobacco leaves were infiltrated with FocCP1 (100 μM), with the same concentrations of Flg22 and BSA as the positive and negative control, respectively. After 24 h treatment, the post-treatment leaves were soaked in 3, 3-diaminobenzidine (DAB)-HCl (1 mg/mL, pH 3.8) solution, and then incubated for 8 h in the dark at room temperature. Subsequently these leaves were immersed into 95% boiling ethanol for 20 min to remove chlorophyll, and then stored in 70% glycerol [65]. H_2_O_2_ production in tobacco leaves was examined under an Olympus Stereomicroscope SZX9 (Olympus, Beijing, China). H_2_O_2_ production in tobacco suspension cell culture was detected by chemiluminescence using luminol as a reagent. The 250 μL cell samples were merged with 300 μL buffer containing 175 mM mannitol, 0.5 mM CaCl_2_, 0.5 mM K_2_SO_4_, and 10 mM HEPES (4-(2-hydroxyethyl)-1-piperazineethanesulfonic acid), pH 5.75. After incubation for 1 h on a rotary shaker (150 rpm) at 26 °C, 20 μL purified FocCP1 (100 μM) and 50 μL luminol (0.3 mM) were added to the solution, with the same concentrations of Flg22 and BSA as the controls. Chemiluminescence was measured using a GloMaxH-96 Luminometer (Promega, Madison, WI, USA) within a 30 s time period. H_2_O_2_ production was expressed as nanomole of H_2_O_2_ per gram fresh weight of tobacco suspended cells [66].

### 4.7. RNA Extraction and Quantitative Real-Time Polymerase Chain Reaction (qRT-PCR)

To analyze whether FocCP1 exhibited elicitor function, 4-week-old tobacco leaves were infiltrated with FocCP1 (100 μM) with BSA (100 μM) as a control. The samples were cryo-preserved in liquid nitrogen post-treatment of 8, 16, and 24 h, three biological replicates were performed [15]. All total RNA was extracted with the RNA Prep-Pure Plant Kit (TianGen, Beijing, China) according to the manufacturer’s protocol. First-strand cDNAs were synthesized with the same method as in Section 4.4. Six defense-related genes, *PR1*, *PR5*, *PAL*, *EDS1*, *HSR203J*, and *LOX*, were amplified using an ABI 7500 Real-Time PCR system (Applied Biosystem, Foster CA, USA) with Trans Start^®^ Green qPCR Super Mix UDG (TransGen Biotech, Beijing, China). Tobacco actin gene was the endogenous control and the qRT-PCR primers are listed in Table 3 [67]. The amplification program was as follows: 50 °C for 2 min, 94 °C for 10 min, followed by 40 cycles of 94 °C for 5 s, 60 °C for 15 s, and 72 °C for 15 s. The relative mRNA expression values were calculated according to 2^−ΔΔCt^ methods [68]. For each gene, qRT-PCR assays were repeated three times with three biological replicates each time.

### 4.8. Quantification of Salicylic Acid (SA) and Jasmonic Acid (JA)

Tobacco leaves were infiltrated with FocCP1 (100 μM) for 24 h with BSA (100 μM) as a control. Leaf samples were ground into power with liquid nitrogen and quantitated for free SA as described previously [69]. About 100 mg tissue was homogenized in 750 μL of methanol-H_2_O-acetic acid (80:19:1, *v*/*v*) with the internal standard (2 μg naphthaleneacetic acid), extracted 16 h or overnight in the dark, and centrifuged at 13,000 rpm for 15 min at 4 °C. The supernatant was collected and the pellet was extracted again with 400 μL buffer without an internal standard for 4 h. Then, all supernatants were combined and filtered with a 0.22 μm nylon filter. 1 mL chloroform was added, samples were centrifuged again and the supernatants was dried by evaporation under nitrogen gas flow at room temperature, then re-dissolved in 200 µL methanol and centrifuged to ensure get out of solid impurities, the supernatants were diluted 100 times with methanol for later use [17,70].

To extract JA in tobacco leaves treated by FocCP1 as above, 10 g tobacco leaves were fully ground, transferred into 50 mL tube and dissolved in 20 mL methanol at 80 °C for 5 min. The samples were vortexed and ultrasonically extracted for 2 h, and then centrifuged at 1000 rpm for 15 min at 4 °C. The supernatant was transferred carefully to a new 50 mL tube and the pellet was extracted again [71]. Then, all supernatants were combined, a little anhydrous sodium sulfate was added, and they were dried by a nitrogen rotary evaporator. The sample was re-dissolved in 2 mL sterile water and purified using NH_2_-HPE purification column (Thermo Scientific, Waltham, MA, USA), the sample was rinsed with 6 mL sterile water, eluted and collected with 2 mL 2% acetic acid in methanol. The JA sample was re-dried and re-dissolved with 1 mL of methanol: water (1:1), and passed through a 0.22 μm filter for later use [72]. About 1 mg of SA or JA standard sample was diluted by methanol, and the standard dilution (1, 5, 10, 50, 100, 500 ng/mL) acted as standard shop to make a standard curve. The content of SA and JA in the final extract was assayed respectively through HSS T3 C18 column (100 × 2.1 mm, 1.7 µM) by HPLC-tandem mass spectrometry (MS/MS) (Thermo Scientific, Waltham, MA, USA) [70,72].

### 4.9. Assays of Defense Responses in Tobacco

To visualize callose deposition, tobacco leaves were infiltrated as above and stained with aniline blue (Aladdin, Shanghai, China) [41]. The steps were as follows: first, the leaves were fixed in solution, which contained 1% (*v*/*v*) glutaraldehyde, 5 mM citric acid and 90 mM Na_2_HPO_4_ (pH 7.4). Then, chlorophyll was removed, and leaves were dehydrated in 100% boiling ethanol. Leaves were then stained in 0.1% (*w*/*v*) aniline blue solution for 1 h (0.1% aniline blue dissolved in 67 mM K_2_HPO_4_, pH 12.0) and subsequently transferred into 50% (*v*/*v*) ethanol, equilibrated in 67 mM K_2_HPO_4_ (pH 12.0), and preserved in glycerol and staining (7:3) solution. At last, the callose was observed on a microscope slide under ultraviolet epifluorescence microscope (Carl Zeiss, Jena, Germany). To detect phenolic compounds accumulation in tobacco suspension cells, 500 μL of tobacco cell suspension was incubated with 100 μL FocCP1 (100 μM) at 120 rpm and 26 °C in the dark for 4 days, the same concentrations of Flg22 and BSA as the controls. Subsequently, the tobacco suspension cells were rinsed thrice with fresh suspension culture buffer [39,41]. The phenolic compounds deposition in tobacco cells were observed under an inverted laser scanning confocal microscope (Zeiss LSM 510, Oberkochen, Germany).

### 4.10. Bioassay for FocCP1-Induced Disease Resistance in Plants

To determine whether FocCP1 induced systemic resistance in tobacco, three lower leaves of tobacco were infiltrated with FocCP1 (100 μM), with the same concentrations of Flg22 and BSA as the controls. Three days later, the upper leaves were inoculated with TMV-GFP virus. Five days later, these leaves were observed under a 265 nm UV lamp (WD-9403E). The number of TMV lesions in each leaf was recorded, and the inhibition rate was calculated using the following formula (**1**) with three replicates [73,74]:
Percent inhibition = [(number of lesions on control leaves-number of lesions on elicitor-treated leaves)/number of lesions on control leaves] × 100%
(1)

*Pst.* 6605 was used to identify tobacco systemic resistance as described in a previous study [75]. Briefly, FocCP1 (100 μM) infiltrated tobacco leaves as above for 2 days, with the same concentrations of Flg22 and BSA as the controls. 50 μL *Pst.* 6605 (OD600 = 0.6) was also infiltrated into tobacco leaves. Four days later, three leaf disc samples were collected from each leaf using a sterilized 1.5 cm diameter punch, and these tissues were thoroughly ground with 100 μL sterile water, the suspension was vortexed and serially diluted to 10^−6^. Dilutions of 100 μL were plated on KB plates and cultured for 2 days, according to leaf disc area (approximately 0.1963 cm^2^); the number of CFU per cm^2^ was calculated by dilution factor [39].

Two-month-old banana seedlings were treated by 100 μM FocCP1 three times using root irrigation every three days; water and BSA (100μM) were the controls. Seven days later, banana seedlings were inoculated with 50 mL 1 × 10^5^ spore concentration FOC4 fermentation broth; these banana seedlings continued to grow for 8 weeks in a greenhouse. According to the leaf yellowing rate and the browning rate of bulbs and vascular bundles, the severity of banana wilt was calculated. Disease index classification criteria of banana blight are listed in Table 4 [76].

## 5. Conclusions

In conclusion, FocCP1 is a small secreted protein with a signal peptide of 18 amino acids and 121 mature amino acids in FOC4, and FocCP1 belongs to the CPP family. This study not only characterized and purified FocCP1 protein but also analyzed the mechanism for triggering the plant immune system. FocCP1 was an elicitor or effector that triggered an immune response and caused SAR in tobacco. Our results provide the possibility for further research on the immune mechanism of plants induced by this protein and lay a research foundation for new biological methods to prevent and control banana wilt in the future.

## Figures and Tables

**Figure 1 ijms-20-02849-f001:**
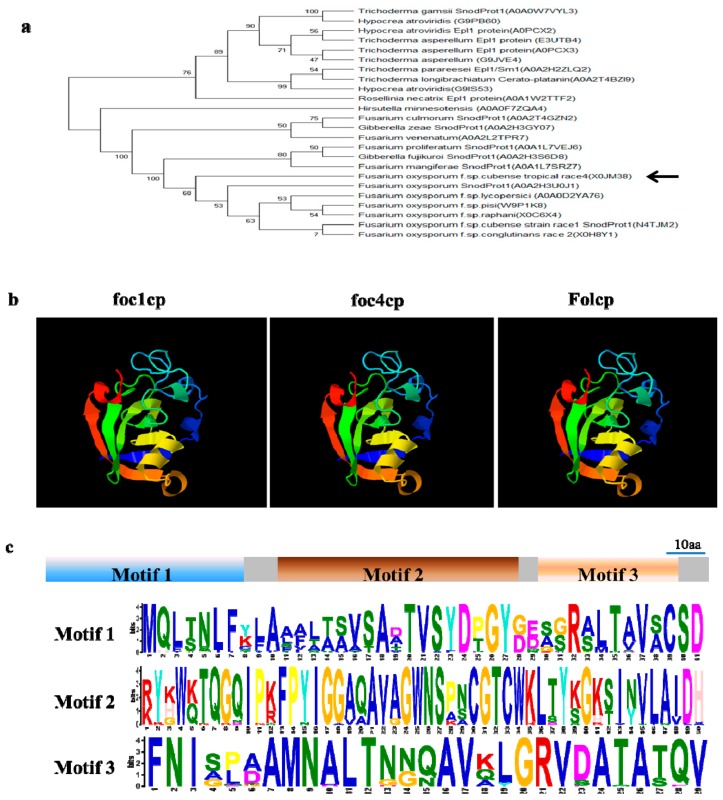
Bioinformatics analysis of the cerato-platanin-like FocCP1 protein. (**a**) The phylogenetic relationship of FocCP1 and other CPPs was determined using the maximum-likelihood algorithm. Branch lengths are proportional to the average probability of change for characters on branch. The arrow represents FocCP1 protein. (**b**) The predicted 3D structures of FocCP1 compared with Foc1CP (N4TJM2) and FolCP (A0A0D2YA76); there is high similarity in the 3D structures of CPPs. (**c**) Conserved motifs of CPPs were predicted using the MEME suite. CPPs exhibit three motifs, and each motif possesses some conserved amino acid. In total, 48 CPPs were used for analysis.

**Figure 2 ijms-20-02849-f002:**
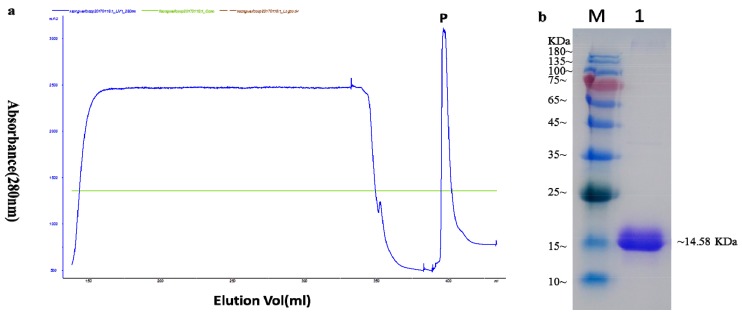
Purification of recombinant FocCP1 protein from *Pichia pastoris* cells via an ÄKTA Protein Purifier System. (**a**) The peak of separation and purification of FocCP1 protein under 280 nm ultraviolet (UV) light. Recombinant FocCP1 protein in fermentation filtrate was collected through a 5 mL His-Trap HP column at a flow rate of 2 mL/min, P: FocCP1 peak. (**b**) Sodium dodecyl sulfate polyacrylamide gel electrophoresis (SDS-PAGE) analysis of the purified recombinant protein. FocCP1 showed a single band with Coomassie Brilliant Blue R-250 staining, M: protein molecular weight marker, 1: FocCP1.

**Figure 3 ijms-20-02849-f003:**
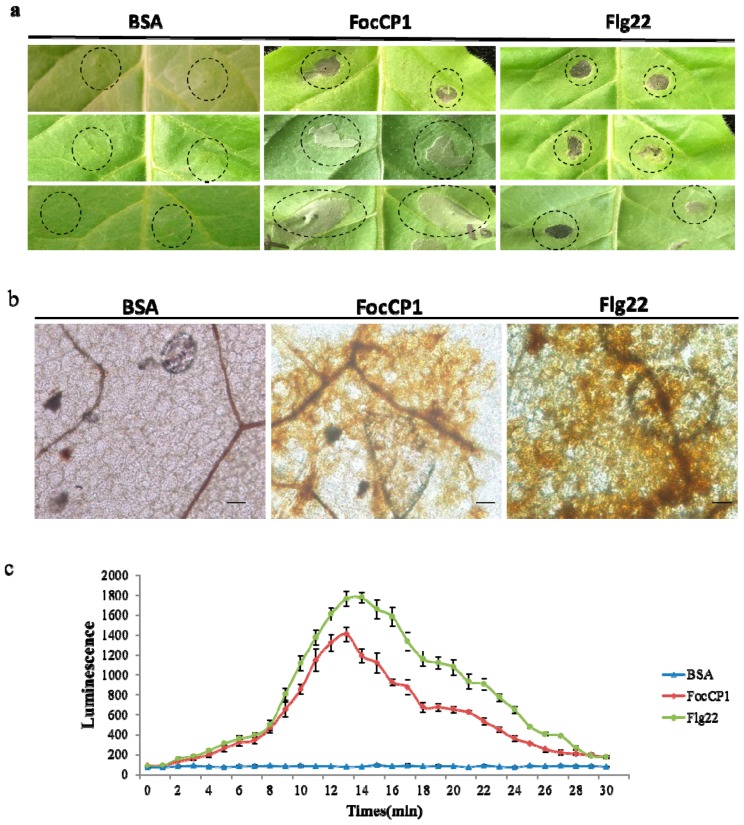
FocCP1 caused hypersensitive response (HR) and reactive oxygen species (ROS) accumulation in tobacco. (**a**) HR was measured in tobacco leaves infiltrated with FocCP1 protein. Six leaves of three plants were used for each replication. (**b**) ROS accumulation was observed in tobacco leaves infiltrated with FocCP1 under microscopy, Flg22 induced obvious H_2_O_2_ accumulation, but bovine serum albumin (BSA) did not. Scale bar = 50 μm. (**c**) ROS burst was detected in tobacco cell culture after FocCP1 treatment, ROS formation improved: better increased after 5 min and reached a maximum at approximately 12–14 min, and gradually lowered to the level of the negative control. In all experiments, flg22 and bovine serum albumin (BSA) were the positive and negative controls, respectively, and three independent replicates were performed. Values are means ± standard error (SE).

**Figure 4 ijms-20-02849-f004:**
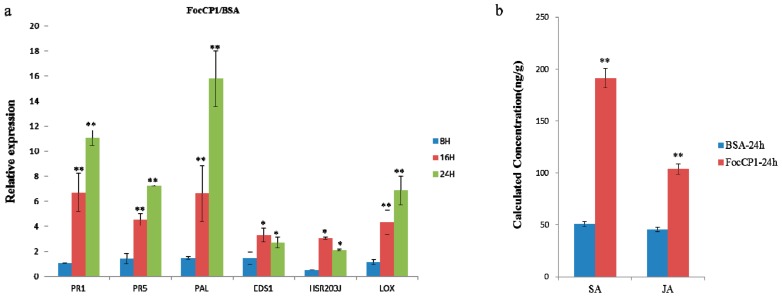
FocCP1 exhibited elicitor function in tobacco. (**a**) The expression of pathogenesis-related *PR* (*PR-1, PR-5*) genes was induction started after 16 h treatment and obviously induction started after 24 h treatment. The marker gene (*PAL*) of the salicylic acid (SA)-dependent defense pathway was significantly induction started by FocCP1 after 24 h treatment, but strong induction was not observed for *EDS1*. The HR marker gene (*HSR203J*) was induced by FocCP1 after 16 h treatment. The same expression pattern was found for the jasmonic acid (JA)/ethylene (Et) pathway gene (*LOX*) after 16 h treatment. All related genes were not significantly up-regulated after 8 h induction. (**b**) The contents of SA and JA were increased in tobacco leaves treated by FocCP1 after 24 h induction. Essentially identical results were obtained in three independent experiments. Values were means ± SE. Statistical analysis of data was conducted using SAS (*, *p* < 0.05; **, *p* < 0.01).

**Figure 5 ijms-20-02849-f005:**
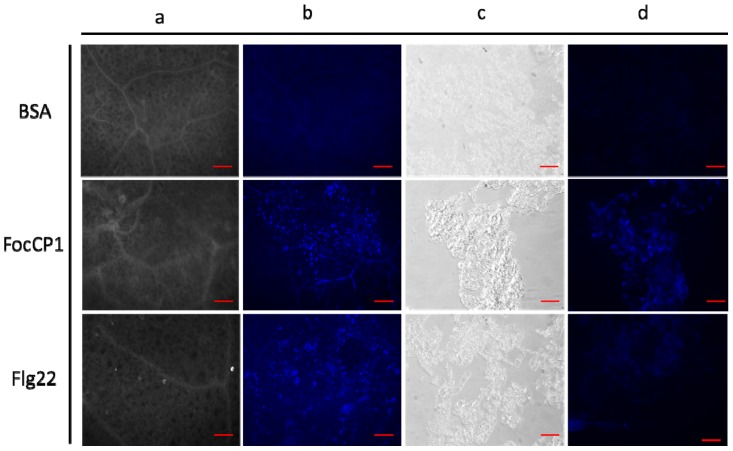
FocCP1 stimulated callose deposition in tobacco leaves and phenolic compounds in tobacco suspension cells. (**a**) Tobacco leaves were observed under bright-field microscopy; (**b**) Tobacco leaves were observed under UV fluorescence microscopy; (**c**) The cell suspension samples were observed under bright-field microscopy; (**d**) The cell suspension samples were observed under UV fluorescence microscopy. The callose deposits and phenolic compounds were photographed in tobacco leaves and suspension cells treated by FocCP1; the positive control Flg22 had an effect on tobacco leaves and suspension cells, while BSA had no effect. Scale bar = 10 μm. For (**a**) and (**b**), six leaves of three plants were used for each replication. For (**c**) and (**d**), three-tube tobacco suspension cells were used. In all experiments, three independent replicates were performed.

**Figure 6 ijms-20-02849-f006:**
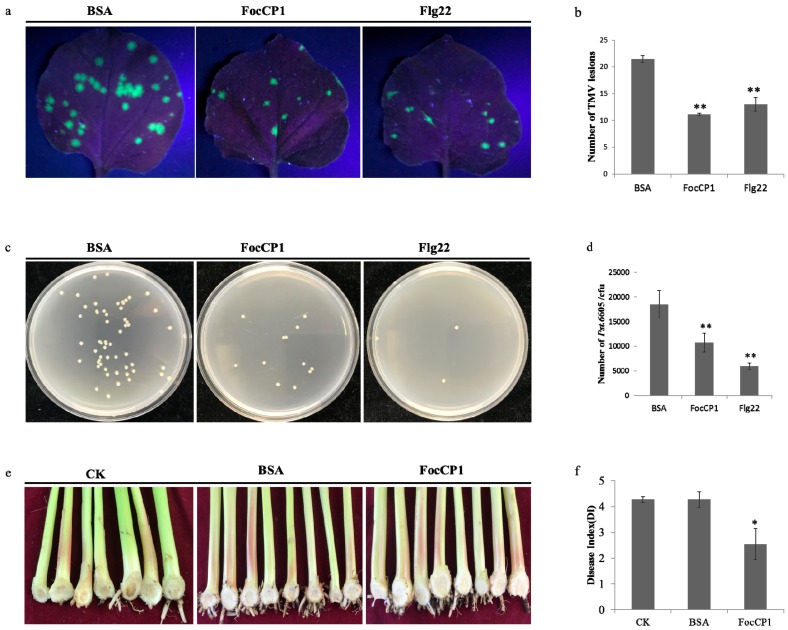
FocCP1 induced systemic acquired resistance in tobacco and enhanced banana seedling resistance against FOC4 infection. (**a**) Tobacco mosaic virus (TMV)-green fluorescent protein (GFP) in tobacco leaves treated by FocCP1 was observed and counted under a UV lamp; BSA and Flg22 were the negative and positive controls, respectively. (**b**) TMV-GFP incidence was calculated in tobacco leaves. TMV lesions in tobacco leaves treated by FocCP1and Flg22 were fewer and smaller than in the control leaves. (**c**) Number of *Pst.* 6605 in tobacco leaves treated by FocCP1 was observed on a King’s B (KB) medium plate, BSA and Flg22 were the controls. (**d**) The concentration of *Pst.* 6605 in tobacco leaves treated by FocCP1 and Flg22 was significantly less than in BSA treated plants. (**e**) Browning in pseudo-stem and bulb of banana plants was counted after inoculation of FOC4 with water as a blank control (CK) and BSA as a negative control. (**f**) The disease index of banana wilt caused by FOC4 after FocCP1 treatment was counted in banana plants. For (**a**) and (**b**): 10 leaves of five plants were used for replication, respectively. For (**c**) and (**d**): six leaves of tobacco plants were used for replication. For (**e**) and (**f**): nine banana plants were used for replication. In all experiments, three independent replicates were performed. Values were means ± SE. Statistical analysis of data was done using SAS (*, *p* < 0.05; **, *p* < 0.01).

**Table 1 ijms-20-02849-t001:** Putative pathogen–host interaction proteins in the secretome.

Accessions ^1^	Protein Name ^2^	Y/T ^3^
*Cellulases*		
N4UR49	Putative glucan endo-1,3-beta-glucosidase	Y
N4U233	Beta-glucosidase	T
N1S288	Glucan endo-1,3-beta-glucosidase	Y
N1RLP0	Putative beta-glucosidase	Y
N1S1N8	Glucan endo-1,6-beta-glucosidase	Y
Chitinase		
N1RY32	Chitinase	Y
N1RJ75	Endochitinase	Y
*Carboxypeptidases*		
X0JJ77	Carboxypeptidase	Y
X0K0U8	Carboxypeptidase	Y
N4UD77	Carboxypeptidase	Y
X0KT75	Glutamate carboxypeptidase	T
N4TK61	Putative metallocarboxypeptidase	Y
X0J1W7	Carboxypeptidase	Y
N1RF81	Putative carboxypeptidase	Y
Lipase		
X0J817	Phospholipase	Y
N1RKZ8	Lysophospholipase	Y
N1RV52	Lipase	Y
*Aminopeptidases*		
X0JV19	Peptide hydrolase	Y
X0JV19	Peptide hydrolase	Y
X0JUF5	Dipeptidyl aminopeptidase	T
N4UIL4	Peptide hydrolase	Y
*Glucanase*		
X0JFN8	Endo-1,3(4)-beta-glucanase	Y
*Transpeptidase*		
N4UHE3	Gamma-glutamyl transpeptidase	Y
*Cerato-platanin protein*		
X0JM38	Uncharacterized protein	Y

^1^ protein ID number from UniProt. ^2^ name and function of the protein. ^3^ whether the protein has a signal P (Y) or trans-membrane domain (T).

**Table 2 ijms-20-02849-t002:** Primers used for polymerase chain reaction (PCR) of *FocCP1* genes.

Genes	Primer Sequences (5′-3′)	Enzyme Cutting Sites
*FocCP1*-F	ATGCAGCTGACCAACCTCTTC	
*FocCP1*-R	TTTGAGACCACAGTTGCTAATAG	
*pPICZαA-FocCP1*-F	CCG**GAATTC**GCGACTGTCTCCTACG ^1^	*EcoR*I
*pPICZαA-FocCP1*-R	GC**TCTAGA***CC*TTTGAGACCACAGTTG ^2^	*Xba*I

^1^ Bold letters: restriction enzyme cutting sites, ^2^ Italicized and underlined letters: frame-shifted bases

**Table 3 ijms-20-02849-t003:** Primers used for quantitative real-time polymerase chain reaction (qRT-PCR) of pathogenesis-related genes.

Genes	Forward Primer (5′-3′)	Reverse Primer (5′-3′)
*NtPR1*	CGTTGAGATGTGGGTCGATG	CCTAGCACATCCAACACGAA
*NtPR5*	CTCATGCTGCCACTTTTGAC	CTCCAAGATTGGCCTGAGTC
*NtPAL*	GTTATGCTCTTAGAACGTCGCCC	CCGTGTAATGCCTTGTTTCTTGA
*NtHSR203J*	TGCCGTCAAAGATGTAGTCG	CAGCATGGCTGACACAAAAG
*NtEDS1*	GGAGAATGGGAGAAGCAGAA	GAACGCATCATAATACCCGA
*NtLOX*	CTTTAAGAGGAGATGGAACT	TCTAAGCTCATAAGCAATGG
*Ntactin*	ATGCCTATGTGGGTGACGAAG	TCTGTTGGCCTTAGGGTTGAG

**Table 4 ijms-20-02849-t004:** Disease index classification criteria.

Disease Stage	Disease Severity Index of Roots/ Bulbs ^1^	Disease Severity Index of Leaves ^2^
0	pseudostems and bulbs were white and healthy	leaves were green, healthy and bright
1	vascular bundle of bulb was sporadically brown	leaves were green, healthy and bright
2	vascular bundle of bulb was sporadically brown, 1/3 of bulb area was yellow	leaves were green, not yellow
3	vascular bundle of bulb was brown, 1/3–2/3 of bulb area was yellow	a few leaves were a little yellow
4	vascular bundle of bulb was brown, more than 2/3 of bulb area was yellow	a few leaves were yellow
5	vascular bundle was brown, 1/2 of the bulb area was brown and decayed	most leaves were yellow
6	vascular bundle was brown, more than 1/2 of bulb area was brown and decayed	leaves were withered

^1^ main standard of disease index, ^2^ secondary standard of disease index

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
