# Peer review of "The Novel Cerato-Platanin-Like Protein FocCP1 from Fusarium oxysporum Triggers an Immune Response in Plants"

_ijms, 2019, doi:10.3390/ijms20112849_

Reviewer 1 Report

The manuscript by Li et al. describes identification and first analysis of a Cerato-plantinin protein in the secreted proteome of Fusarium oxysporum f.sp. cubense (strain FOC4). The plant response to the heterologous expressed and purified protein FocCP1 was tested in tobacco leaves and tobacco suspension cells. Different methods were used to show a plant defence eliciting function of the purified protein, as compared to Flg22 (positive control) and BSA (negative control) treatment. Triggered plant disease resistance after treatment with FocCP1 against TMV and Psa was shown in tobacco and against FOC4 in banana seedlings.

 Overall the data are nicely presented and the manuscript is well written. However, the manuscript would benefit from engl. proof reading, with special attention on use of tense and singular/ plural.

General:

Results shown in Figure S3 and S4 should be mentioned briefly in the results section.

Please start each results section with an introductory sentence, as done in 2.4, 2.5 and 2.6.

Line 159 and elsewhere: The word excitation seems not an appropriate term. Please use: elicit, evoke, cause, induce, provoke etc. instead.

Some details are missing in the M and M section:

Line 326: how was the induction culture performed in detail?

Line 369 and elsewhere: What was the concentration of flg22 and BSA used in the different experiments?

Figures:

Labelling of sub-images with (a), (b) etc. better place in left, upper corner of image.

Scale bar in fig. 5 is missing or hard to see. The images under a. are very dark, especially when printed. Same for d. Fluorescence is hardly visible.

Discussion

The interpretation of results is in many cases very strong. Especially results inferred from bioinformatics analysis give just indications of biological functions and structures. E. g. a signal peptide indicates that the protein is secreted but this should be confirmed experimentally. The 3D reconstruction indicates a certain structure but does not prove it.

Please speculate on the SA and JA results. Usually these pathways act antagonistically (see e.g. Ning Li et al 2019 Int. J. Mol. Sci)

Line 310: Please give some ideas on how the interaction mechanism of elicitor and plant could be further analysed.

Line 459: Can you speculate on how the proteomics results can be used to control banana wilt disease in future?

Can you speculate, how the immune response results can be transferred from tobacco to the banana root system? Have similar studies been done in banana?

   Similar assays as performed with tobacco would be interesting on the host plant banana- Do you see HR/ Callose/ Phenolic compound responses here.

Minor:

39:   comma missing before, respectively

71/73: reword PAMP posesses

77: NB-LRR: add nucleotide

78: leucine-rich

79: ...associated with production of ROS and subsequent HR

81: of pathogenic microorganisms

90. the results of the study provide.... and lays a

100: are listed in Figure S...

131: The recombinant....

166: improved: better increased after 5 min

187: Please check this elsewhere as well: gene induction started 16 h after treatment / induction

196: ROS is

201: ...Flg22 treatment resulted...

202: ...components are considered

203: ...structure and act against...

220: ...less than in BSA treated plants and...

234:...smaller than in...

238:.... plants was counted...

247: ..potentially secreted

248: may play or most likely play – an important role for secreted this was not shown in the manuscript

254: The proteomics and bioinformatics analysis indicated that FocCP1 is a ...

258: protein excitation, better use e.g. elicitor function

273: Can you speculate on the abundance / concentration of FocCP1 during banana infection?

286: We speculate...involves...

287: the HSR gene was consistent...do you mean upregulation of expression of the gene was consistent

304: we speculate...inhibits...

 Author Response

Response to Reviewer 1 Comments

 Comments and Suggestions for Authors and Responses:

Point 1:  General:  Results shown in Figure S3 and S4 should be mentioned briefly in the results section.

ResponseOk, these results will be mentioned briefly in the results section.

Point 2:  Please start each results section with an introductory sentence, as done in 2.4, 2.5 and 2.6.

ResponseThanks for reviewer’s comment. We have re-written these sentences according to the reviewer’s suggestion.

Point 3: Line 159 and elsewhere: The word excitation seems not an appropriate term. Please use: elicit, evoke, cause, induce, provoke etc. instead.

ResponseThanks for reviewer’s comment. We have changed this word according to the comment.

Point 4: Some details are missing in the M and M section:

Line 326: how was the induction culture performed in detail?

ResponseWe have changed the following sentence, “…. Subsequently, FOC4 fermentation liquid was induction cultured with sterile banana seedlings for 2 days” into “FOC4 fermentation liquid continued to be co-cultivated under inducing by the hung sterilized banana seedlings for 2 days.”

Point 5: Line 369 and elsewhere: What was the concentration of flg22 and BSA used in the different experiments?

ResponseWe have added the concentration of Flg22 and BSA in the different experiments.

Point 6: Figures: Labelling of sub-images with (a), (b) etc. better place in left, upper corner of image.

ResponseThanks for reviewer’s comment. We have modified it according to the reviewer’s suggestion.

Point 7: Scale bar in fig. 5 is missing or hard to see. The images under a. are very dark, especially when printed. Same for d. Fluorescence is hardly visible.

ResponseThanks very much for this comment. We have modified Fig 5but, all pictures are dark under UV fluorescence microscopy, we can only adjusted the brightness of these images. We have done it.

Point 8: Discussion: The interpretation of results is in many cases very strong. Especially results inferred from bioinformatics analysis give just indications of biological functions and structures. E. g. a signal peptide indicates that the protein is secreted but this should be confirmed experimentally. The 3D reconstruction indicates a certain structure but does not prove it.

ResponseA signal peptide indicates that FocCP1 may be a secreted protein, Label-free analysis of FOC4 strain showed that the FocCP1 was found in fermentation broth, and the 3D reconstruction indicates that the FocCP1 has a similar structure to other CPPs, CPPs are considered to be secreted protein. This paragraph is only a bioinformatics analysis.

Point 9: Please speculate on the SA and JA results. Usually these pathways act antagonistically (see e.g. Ning Li et al 2019 Int. J. Mol. Sci)

ResponseYes, indeed, we know the SA and JA/Et pathway act antagonistically. The SA-mediated defense response plays a central role in local and systemic-acquired resistance (SAR) against biotrophic pathogens, while the JA/Et-mediated response contributes to the defense against necrotrophic pathogens.

  We speculate that elicitor protein owns similar induction mechanism with biotrophic pathogens and necrotrophic pathogens. When the tobacco leaves were infiltrated with FocCP1, the FocCP1 protein firstly triggered SA pathway and induced SAR in tobacco, and the FocCP1 protein in the cytoplasm also triggered JA/Et pathway and caused HR in tobacco leaves. We just determined the content of SA and JA in tobacco leaves treated by FocCP1 protein after 24 h treatment. More experiments should be used to prove the speculated conclusion.

Point 10: Line 310: Please give some ideas on how the interaction mechanism of elicitor and plant could be further analysed.

ResponseThere are different immune response stages in plants induced by protein elicitors. Firstly, the protein elicitors are monitored and identified by membrane-anchored pattern recognition receptors (PRRs) in plant cell membranes, triggered pathogen-associated molecular pattern (PAMP)-triggered immunity (PTI) and caused local and systemic-acquired resistance (SAR) in plants (Chao Yang, Jun Liu, etc. Binding of the Magnaporthe oryzae chitinase MoChia1 by a rice tetratricopeptide repeat protein allows free chitin to trigger immune responses. plant cell advance publication. 1,2019. doi:10.1105/tpc.18.00382). Secondly, the peptides or fragments of protein elicitor are endocytosed into plant cells (In the following experiments, we found the related genes involves in endocytosis were regulated in tobacco sprayed by protein elicitor), so that, the protein elicitors are identified by R proteins in plant cell and trigger the strong immune responses, such as, reactive oxygen species (ROS) and programmed cell death (PCD).

Point 11: Line 459: Can you speculate on how the proteomics results can be used to control banana wilt disease in future?

ResponseMore protein elicitors will be found in the proteomics of FOC4 fermentation broth, these elicitors trigger banana seedlings immune resistance against banana wilt infection, the protein elicitors are acted as an immune inducer substances to the culture nutrient matrix for banana seedlings.

Through compared proteomics in FOC4 fermentation broth and FOC4 fermentation broth induced by plant seedlings, we should detect some proteins associated with pathogenesis and further research on the gene functions to enhance the ability of banana plant to resist FOC infection.

Point 12: Can you speculate, how the immune response results can be transferred from tobacco to the banana root system? Have similar studies been done in banana?

Similar assays as performed with tobacco would be interesting on the host plant banana- Do you see HR/ Callose/ Phenolic compound responses here.

ResponseMore experiments will be carried out, the expression of pathogenesis-related defense genes, defense gene-related pathways, and hormone-related pathways will be determined in banana plant treated by FocCP1 elicitor, the early response events in banana, including oxygen bursts and accumulation of phenolic compounds, will be proved.  Because the leaves of banana are not easily injected, more experiments will be performed in suspension cells and banana roots.

Minor:

Point 13:

L. 39:   comma missing before, respectively

L.71/73: reword PAMP possesses

L.77: NB-LRR: add nucleotide

L.78: leucine-rich

L.79: ...associated with production of ROS and subsequent HR

L.81: of pathogenic microorganisms

L.90. the results of the study provide.... and lays a

L.100: are listed in Figure S...

L.131: The recombinant....

L.166: improved: better increased after 5 min

L.187: Please check this elsewhere as well: gene induction started 16 h after treatment / induction

L.196: ROS is

L.201: ...Flg22 treatment resulted...

L.202: ...components are considered

L.203: ...structure and act against...

L.220: ...less than in BSA treated plants and...

L.234:...smaller than in...

L.238:.... plants was counted...

L.247: ..potentially secreted

L.248: may play or most likely play – an important role for secreted this was not shown in the manuscript

L.254: The proteomics and bioinformatics analysis indicated that FocCP1 is a ...

L.258: protein excitation, better use e.g. elicitor function

ResponseYes, thank you very much for these comments, we have re-written these sentences mentioned above according to the reviewer’s suggestions.

Point 14:

L.273: Can you speculate on the abundance / concentration of FocCP1 during banana infection?

ResponseThank you very much for this suggestion. But we have not measured the concentration of FocCP1 infecting bananas. However, we speculate that the content of FocCP1 might be very low. Subsequent experiments should be designed.

Point 14: L.286: We speculate...involves...

ResponseThanks for reviewer’s comment. We have reorganized the sentence.

Point 15: L.287: the HSR gene was consistent...do you mean upregulation of expression of the gene was consistent

ResponseFor better understanding, we have modified the sentence into: “In addition, the HSR203J gene was involved in chitinase synthesis and HR symptoms in plant”

Point 16: L.304: we speculate...inhibits...

ResponseThanks for reviewer’s comment. We have reorganized the sentence.

Reviewer 2 Report

The article by Li et al. is very nicely written, it flows well and is well structured. My only general comment would be about the conclusion of the authors that the "protein FocCP1 in inducing SAR in plants.".

I don’t agree that the authors can generalize already their results to any plant species, I would recommend to only write “cause SAR in tobacco” (l. 457). Their results only suggest a similar action in banana seedlings, but further analyses are needed to conclude for the same activity in banana. The only infection test done with banana seedlings does not allow generalization.

Several points need to be clarified in the Material and Method section (see below).

I recommend the manuscript to be accepted after minor revisions according to my comments below.

Fig 5 is not visible (except panel c)

l. 301: The efficiency of FocCP1 on banana presented in this study remains light and should be described only as “suggested”.

l. 326: Are there words missing in the sentence?

l. 330: which protease inhibitor mixture has been used?

l. 360: the FocCP1 fusion protein contains the HIS tag. Could the author comment about the possible role of the HIS tag in the activity of the protein?

l. 367: How were the leaves infiltrated with the FocCP1solution ?

l. 385: Is it 200 and 350 µl instead of ml? So is it right that the final FocCP1 concentration was 161µM in the well?

l. 270, 369: Where do the authors got the Flg22 protein? Any reference about it?

l. 411. Extraction: I understand that only free SA was quantified, and then only the organic phase was analyzed..? The authors should complete the method here.

l. 415: JA analysis: none of the articles cited (#19, 72, 74) describe the JA extraction and analysis. The authors should add their own method (LC-MS conditions) if it has not been described previously.

l. 457: “cause SAR in plants”. please see my general comment above.

l. 479: the necrotic area does not mean much here as the infiltrated area is not mentioned. Should it be the proportion of the infiltrated area that became necrotic?

Author Response

Response to Reviewer 2 Comments

 Comments and Suggestions for Authors

The article by Li et al. is very nicely written; it flows well and is well structured. My only general comment would be about the conclusion of the authors that the "protein FocCP1 in inducing SAR in plants."

Point 1: I don’t agree that the authors can generalize already their results to any plant species; I would recommend to only write “cause SAR in tobacco” (l. 457). Their results only suggest a similar action in banana seedlings, but further analyses are needed to conclude for the same activity in banana. The only infection test done with banana seedlings does not allow generalization.

Response: Thanks very much for the comment. We agree with this suggestion and have modified “inducing SAR in plants” into “inducing SAR in tobacco”.

Point 2: Several points need to be clarified in the Material and Method section (see below). I recommend the manuscript to be accepted after minor revisions according to my comments below.

Fig 5 is not visible (except panel c)

Response: Thanks very much for this comment. We have modified Fig 5but, all pictures are dark under UV fluorescence microscopy, we can only adjusted the brightness of these images. We have done it.

Point 3: L. 301: The efficiency of FocCP1 on banana presented in this study remains light and should be described only as “suggested”.

Response: Thanks for reviewer’s comment. We have changed the sentenceMoreover, FocCP1 triggered tobacco resistance to TMV and Pst. 6605 and improved the ability of banana seedlings to resist FOC4 infection (Figure 6)” into “Moreover, FocCP1 triggered tobacco resistance to TMV and Pst. 6605 and reduced the probability of Fusarium oxysporum infesting banana seedlings (Figure 6)”. Add a sentence “More experiments should be proved that the FocCP1 induced SAR in banan”.

 Point 4: L. 326: Are there words missing in the sentence?

Response: There were no words missing in the sentence. In order to understand the sentence better, we changed the sentence “FOC4 fermentation liquid was induction cultured with sterile banana seedlings for 2 days” into “Subsequently, FOC4 fermentation liquid continued to be co-cultivated under inducing by hung sterilized banana seedlings for 2 days”.

Point 5: L. 330: which protease inhibitor mixture has been used?

Response: The protease inhibitor was 1 mM PMSF solution; we have added it in the sentence.

Point 6: L. 360: the FocCP1 fusion protein contains the HIS tag. Could the author comment about the possible role of the HIS tag in the activity of the protein?

Response: The HIS tag was used to isolated and purified the FocCP1 fusion protein, The HIS tag protein could not induce the HR response in tobacco in previous research (such as: Zhang, Y., et al., The Verticillium dahlia SnodProt1-Like Protein VdCP1 Contributes to Virulence and Triggers the Plant Immune System. Frontiers in Plant Science, 2017.8:1880.)

Point 7: L. 367: How were the leaves infiltrated with the FocCP1 solution?

Response: We modified the sentence: “50μl FocCP1 (100 μM) was infiltrated into 4-week-old tobacco leaves with a sterile 1 mL syringe, Bovine serum albumin (BSA) and Flg22 with the same concentration as the negative and positive controls”.

Point 8: L. 385: Is it 200 and 350 µl instead of ml? So is it right that the final FocCP1 concentration was 161µM in the well?

Response: We have modified “ml” to “µl” in the sentence. But 20 µl MoHrip1 (5 µM) and 50 µl of 0.3 µM luminol were put into buffer. (Reference: Chen, M., et al., Purification and characterization of a novel hypersensitive response-inducing elicitor from Magnaporthe oryzae that triggers defense response in rice. PLoS One, 2012. 7(5): p. e37654.).

Point 9:L. 270, 369: Where do the authors got the Flg22 protein? Any reference about it?

Response: We got the BSA (Sigma, 1 mg/ml) and Flg22 protein (Phytotech, 1 mg/ml) from a biochemical reagent manufacturer (Reference: Chen, M., et al., Purification and characterization of a novel hypersensitive response-inducing elicitor from Magnaporthe oryzae that triggers defense response in rice. PLoS One, 2012. 7(5): p. e37654.).

Point 10: L. 411. Extraction: I understand that only free SA was quantified, and then only the organic phase was analyzed..? The authors should complete the method here.

Response: Only free SA was quantified, the content of SA in tobacco was extracted as described previously. (Guogen Yang, Liguang Tang, Yingdi Gong. A cerato-platanin protein SsCP1 targets plant PR1 and contributes to virulence of Sclerotinia sclerotiorum. 2017, 9, doi: 10.1111/nph.14842).  PLS refers to this literature.

 Point 11: L. 415: JA analysis: none of the articles cited (#19, 72, 74) describe the JA extraction and analysis. The authors should add their own method (LC-MS conditions) if it has not been described previously.

Response: We have added the methods: “To extract JA in tobacco leaves treated by FocCP1 as above, 10 g tobacco leaves were fully ground, transferred into 50 ml tube and dissolved in 20 ml methanol at 80°C for 5 min. The samples were vortexed and ultrasonically extracted for 2 h, and then centrifuged at 1000 × g for 15 min at 4°C. The supernatant was transferred carefully to a new 50 ml tube and the pellet was extracted again [74]. Then, all supernatants were combined, added a little anhydrous sodium sulfate and dried by a nitrogen rotary evaporator. The sample was re-dissolved in 2 ml sterile water and purified using NH2-HPE purification column (Thermo Scientific), the sample was rinsed with 6 ml sterile water, eluted and collected with 2 ml 2% acetic acid in methanol The JA sample was re-dried and re-dissolved with 1 ml of methanol: water (1:1), and passed through a 0.22 μm filter for later use[75]. About 1 mg of SA or JA standard sample was diluted by methanol, and the standard dilution (1, 5, 10, 50, 100, 500 ng/ml) acted as standard shop to make a standard curve. The content of SA and JA in the final extract was assayed respectively through HSS T3 C18 column (100× 2.1 mm, 1.7 µM) by HPLC-MS/MS”.

The extraction of salicylic acid and jasmonic acid is carried out by Baimaike Company.

Point 12: L. 457: “cause SAR in plants”. please see my general comment above.

Response: Thanks for reviewer’s comment. We have modified the sentence according to this comment.

Point 13: L. 479: the necrotic area does not mean much here as the infiltrated area is not mentioned. Should it be the proportion of the infiltrated area that became necrotic?

Response: We determined whether FocCP1 protein induced tobacco HR in tobacco, the tobacco leaves were infiltrated with 50 µl FocCP1 (100 µM). The infiltrated area was approximately 2 cm2. We observed that the HR area in tobacco treated by elicitor, the necrotic area was accounted with average calculation.